# Extraction, Structural Characterization, Biological Functions, and Application of Rice Bran Polysaccharides: A Review

**DOI:** 10.3390/foods12030639

**Published:** 2023-02-02

**Authors:** Bingjie Chen, Yongjin Qiao, Xiao Wang, Yi Zhang, Linglin Fu

**Affiliations:** 1School of Food Science and Biotechnology, Zhejiang Gongshang University, Hangzhou 310018, China; 2Institute of Crop Breeding and Cultivation, Shanghai Academy of Agricultural Science, Shanghai 201403, China

**Keywords:** rice bran polysaccharides, structural characterization, biological functions

## Abstract

Rice bran is a “treasure house of natural nutrition”. Even so, utilization of rice bran is often ignored, and this has resulted in the wastage of nutrients. Polysaccharides are one of the active substances in rice bran that have gained widespread attention for their antioxidant, antitumor, immune-enhancing, antibacterial, and hypoglycemic properties. This review summarizes the extraction methods, structural characterization, bioactivity, and application of rice bran polysaccharides that have been developed and studied in recent years, laying a foundation for its development into foods and medicines. In addition, we also discuss the prospects for future research on rice bran polysaccharides.

## 1. Introduction

Rice is a staple food for the majority of the population worldwide, especially in Asia, South America, and Africa. Rice bran, present on the surface of the rice, is the primary by-product of rice, and is separated from the rice grain during the milling of brown rice. It is composed of exocarp, mesocarp, a cross-linked layer, an aleurone layer, and other structural layers, which account for 5–8% of the total rice weight [1] (Figure 1). Currently, global rice production is about 500 million tons each year, generating >25 million tons of rice bran [2]. Rice bran is not only a rich resource, but also has extremely high nutritional value. Rice bran contains 12–16% protein, 12–23% fat, and 23–30% dietary fiber. In addition, it is abundant in vitamins, minerals, and other nutritional factors such as phenanthrene, inositol, phytic acid, and glutamate [3]. Rice bran also nourishes the intestines, increases appetite, tonifies qi and blood, and could be used to treat foot diseases as per ancient Chinese medical texts [4]. Thus, it is regarded as a “treasure house of natural nutrition.” However, currently, the comprehensive utilization rate of rice bran is very low, and most of it is either used as poultry feed or discarded directly, resulting in a significant loss of resources [5]. The United Nations Industrial Development Organization (UNIDO) classifies rice bran as an underutilized natural raw material.

Recent studies have shown that polysaccharides, which are one of the important functional components of rice bran, have anti-tumor [6], antioxidant [7], and immunomodulatory functions [8]. Rice bran polysaccharide also could be used for drug and gene delivery [9]. However, until now, only a single review article on a rice bran arabinoxylan compound (RBAC) modified with shiitake mushroom enzymes is available [10]. The lack of systematic research studies on rice bran polysaccharides (RBP) and the underutilization of RBP have seriously hampered the effective development of RBP. Thus, in this review, we have systematically summarized the research on RBP in the past decades to provide comprehensive insights into the extraction and purification techniques for RBP, as well as its structural characterization and pharmacological activities, with special emphasis on the study of RBP in gene delivery. In addition, we also focused on the food, industrial, and pharmaceutical applications of RBP. The objective of this review is to provide a comprehensive understanding of the research status of RBP and to highlight the great potential of RBP in practical application, so as to realize the full utilization of RBP resources.

## 2. Extraction, Isolation, and Purification of Rice Bran Polysaccharides

### 2.1. Extraction of Polysaccharides

The basic principle of RBP extraction is to disrupt and degrade the cell wall of rice bran using a suitable method to facilitate the release of polysaccharides [11]. Polysaccharides are polar polymer compounds, which are known to be soluble in water and insoluble in alcohols, ethers, acetone, and other organic solvents as per the similarity–solubility principle. Thus, they can be extracted using an aqueous solution [12]. Hot water extraction (HWE) is the most widely used method for polysaccharide extraction due to its ease of process and cost-effectiveness. However, it also has certain disadvantages, such as long extraction time, relatively high required temperature (95–100 °C), and low yield [13]. Thus, the method of polysaccharide extraction needs to be optimized. An orthogonal test was used for increasing the polysaccharide extraction rate and yield, and the maximum polysaccharide yield was 2.18% under the optimal reaction conditions (a solid–liquid ratio of 1:10, at 100 °C for 5 h) [3].

Moreover, microwave, ultrasonic, and enzyme treatments have become excellent substitutes for commonly used techniques due to their high efficiency and environmental friendliness [14]. Han et al. [15] showed that the yield of RBP, which was extracted using the microwave-assisted method (RBP-M) and enzyme-assisted method (RBP-E), was 2.72% and 1.91%, higher than that extracted using hot water (RBP-H, 1.52%). Enzyme-assisted extraction is simple to operate, has mild reaction conditions, and does not change the molecular structure of polysaccharides; thus, it has become a research hotspot [16]. The microwave-assisted extraction method has certain advantages, such as low solvent consumption, fast extraction speed, and no pollution. However, the rapid temperature peak that occurs with the microwave alters the microscopic, chemical, and molecular structure of polysaccharides [17,18]. According to the study of Surin et al. [19], the yield of polysaccharides using HWE (0.8%) was significantly lower than that of ultrasound-assisted extraction (4.0%) under the optimal conditions: 150 W ultrasonic power, 70 °C ultrasonic temperature, and 20 min extraction time. In another study, in the optimized rice bran extrusion conditions, the extraction rate of rice bran polysaccharides was further optimized. The result of this study indicated that under the most favorable conditions—117 W ultrasonic power, 72 °C ultrasonic temperature, and extraction time of 21 min—the extraction rate of polysaccharides could be as high as 7.18% [20]. Polysaccharides can be extracted quickly and efficiently by the cavitation effect of high-frequency ultrasound oscillation, but the molecular weight and structure of polysaccharides could be destroyed by ultrasound waves under the mechanical action, cavitation, and thermal effects. Therefore, the power and time of the ultrasonic waves should be controlled to reduce the degradation of polysaccharides [21,22].

Three-phase partitioning (TPP) is a new method for extraction, purification, and concentration of macromolecules that utilizes different phases formed from raw materials containing solid salts and organic solvents [23]. Wang et al. [24] found that rice bran oil, protein, and RBP could be extracted using (NH4)_2_SO_4_ and tert-butanol. The extraction rate of RBP is around 2.09%, validating the fact that TPP greatly improved the utilization of nutrients. Compared with traditional techniques, TPP has the advantages of solvent recycling, short extraction time, and operation at room temperature, and it could be used as a new technique for the extraction of polysaccharides.

### 2.2. Isolation and Purification of Polysaccharides

Effective separation and purification of polysaccharides were the prerequisites for this study, which aimed to investigate the structure identification and conformational relationship [25]. To extract polysaccharides with high purity, rice brans are usually subjected to a series of separation and purification processes. An organic solvent (petroleum ether) is used for degreasing, and alcohol is used to remove low-molecular-weight impurities in the pretreatment stage [26]. Since the polysaccharides extracted using water extraction and alcohol deposition often contain starch, proteins, and small molecular materials, more appropriate methods are desirable to remove these impurities. Previous studies have shown that the rice brans contain a large quantity of starch, the values of which range from 10% to 55% [27]. Treatment of extracts or fractions with α-amylase could be carried out to remove the starch from the polysaccharides [15,28]. Rice bran is also rich in proteins, and these proteins could be removed using sevag (4:1 v/v chloroform: n-butyl alcohol) [29]. After dialysis, the polysaccharides were further purified using anion-exchange chromatography (DEAE-Cellulose [30], DEAE-Sepharose [31], etc.) and Sephadex gel-permeation [32] depending on the molecular charge and molecular size of the polysaccharides, respectively. The extraction, separation, and purification processes for rice bran polysaccharides are depicted in Figure 2.

## 3. Chemical Characterization of Rice Bran Polysaccharides

The physiological activity of polysaccharides is closely related to their structure. Thus, the structural information of polysaccharides, such as molecular weight, monosaccharide composition, glycosidic bond type, and glycosidic bond conformation, is often used to characterize them [33]. Detailed information on the structure and detection methods of rice bran polysaccharides is summarized in Table 1.

### 3.1. Monosaccharide Composition

As is widely known, polysaccharides are sugar polymeric carbohydrates consisting of at least ten monosaccharides bound by glycosidic bonds. The composition of monosaccharides is usually determined by complete acid hydrolysis and derivatization methods, and detected using high-performance liquid chromatography (HPLC) and gas chromatography-mass spectrometry (GC-MS) [38,39]. Rice bran polysaccharides are a type of heteropolysaccharide composed of galactose (Gal), glucose (Glu), xylose (Xyl), arabinose (Ara), mannose (Man), and rhamnose (Rha) in different ratios [40]. A plethora of studies have explained that composition of heteropolysaccharides in rice bran polysaccharides varies with the rice’s different varieties, extraction methods, and separation procedures. For instance, Li et al. [32] performed gas chromatography and concluded that two polysaccharides, M-50-A and A-50-G, isolated from fermented rice bran consisted of Ara, Xyl, Glu, Man, and Gal in molar ratios of 1.7:7.2:1:11:1.4, and 1.2:1.5:1:11.1:2.4, respectively. Hefnawy et al. [34] isolated three polysaccharides (RBP1, RBP2, and RBP3) from rice bran and found that RBP1 and RBP2 contained Ara, Xyl, Man, Glu, and Gal in molar ratios of 1.00:2.82:57.11: 140.82:7.76 and 1.00:1.62:1.18:77.5:7.79, respectively, whereas RBP3 consisted of Ara, Man, Glu, and Gal in a molar ratio of 1.00:1.03:8.84:2.00.

### 3.2. Molecular Weight Analysis

The molecular weight (Mw) of polysaccharides is usually determined using gel permeation chromatography (GPC), HPLC, and high-performance gel permeation chromatography (HPGPC). Under different experimental conditions, the molecular weight of RBP ranged from 6.95 to 488 kDa (Table 1). Different extraction and purification methods might affect the Mw of RBP. After separation and purification, the average Mw of three rice bran polysaccharide fractions, RBP1, RBP2, and RBP3, were 51.5, 26.1, and 6.95 kDa, respectively [34]. The molecular weights of M-50-A and A-50-G, extracted by Li et al. [32] using different fermentation methods, were 110 and 14 kDa. The Mw of RBP-H, RBP-M, and RBP-E, extracted using the hot water, microwave-assisted, and enzyme-assisted methods, were 1.03 × 10^5^, 2.62 × 10^5^, and 0.46 × 10^5^ g/mol, respectively, as per the analysis using high-performance size-exclusion chromatography and multiangle laser-light scattering using refractive index detection [15]. Therefore, the isolation and purification methods strongly affected the Mw of the polysaccharides.

### 3.3. Chemical Structures

The functional groups are usually detected using Fourier transform infrared spectroscopy (FT-IR), and the linkage type of glycosyl residues is generally determined using a combination of methylation analysis and GC-MS techniques [41]. In a study by Shibuya, Yamagishi, et al. [42,43], two alkali-soluble rice bran polysaccharides were prepared using sodium hydroxide solution. One was highly differentiated arabinoxylan containing xylose residues, and the other one was α-1,4 of glucan, containing glucose and xylose residues. The backbone of the heteropolysaccharide isolated from the defatted rice bran consisted of β-(1 → 3)-linked D-galacopyranosyl residues substituted at O-2 with glycosyl residues composed of α-D-xylose-(1 → 4)-α-D-arabinose-(1 → and α-D-glucose-(1 → 4)-α-D-arabinose-(1 → linked residues [35]. A recent report by Chen et al. [36] demonstrated that rice bran polysaccharides are linked by two glycosidic bonds, α-1,4 and α-1,6, of which α-1,4 has a bigger structure. According to the current findings, the structures of pumpkin polysaccharides have certain regularity. However, elucidation of this regularity requires more studies related to the structure of rice bran polysaccharides.

## 4. Pharmacological Effects of Rice Bran Polysaccharides

The biological activities of rice bran polysaccharides, such as antioxidant, antitumor, immune-enhancing, and hypolipidemic effects, are summarized in Figure 3, which provides a theoretical basis for the application of polysaccharides to treat diseases or as dietary supplements [13].

### 4.1. Antioxidant Activity

In the past few decades, due to the recognition of the impact of oxygen free radicals on cardiovascular diseases, neurodegenerative diseases, aging, and cancer processes, antioxidant activity in food and diet has gained much attention [44]. Multitudinous in vitro studies have confirmed that rice bran polysaccharides have high antioxidant properties, mainly due to the scavenging effect of free radicals, such as hydroxyl (OH•), 1,1-diphenyl-2-picrylhydrazyl (DPPH), and superoxide radical (O_2_^•−^) [13,19,23]. Previous studies have shown that the antioxidant activity of polysaccharides differed when precipitated using different concentrations of ethanol. The three fractions of rice bran polysaccharides, designated as PW1, PW2, and PW3, were precipitated using 40% ethanol, 60% ethanol, and 80% ethanol, respectively. Analyses of these fractions revealed that PW1 exhibited high O_2_^•−^, OH• scavenging ability, and anti-lipid peroxidation at 1.0 mg/mL, while PW3 exhibited high reducing and chelating activities for the formation of ferrous ions [45]. Different extraction methods affect the antioxidant activities of rice bran polysaccharides. Han et al. [14] showed that the enzyme-assisted extracted polysaccharide (RBP-E) had a greater antioxidant effect than the microwave-assisted extracted polysaccharide (RBP-M) at the same concentration in in vivo experiments. Similarly, Zha et al. [7] showed that polysaccharides extracted using water (PW) and 1% (*w*/*v*) NaOH (PN1) had higher antioxidant potential than those extracted with 5% (*w*/*v*) NaOH (PN5). When the concentration of polysaccharides was over 0.075 mg/mL, the total antioxidant activity of the samples decreased in the order of PN1 > PW > PN5. A plethora of studies have shown that the molecular mass, along with the structure of polysaccharides, has a significant influence on biological activity [46]. Experts have previously strived to modify RBP’s physical, chemical, and biological methods to improve its biological activity [47]. Huang et al. [3] chemically modified RBP using carboxymethylated, phosphorylated, and acetylated methods. The result of this analysis showed that the antioxidant activity of phosphorylated RBP was enhanced, while that of acetylated RBP declined. Evaluation of the antioxidant activities of RBP and of the RBP–metal complexes RBP-Fe (III), RBP-Cu, RBP-Zn, and RBP-Ca showed that RBP-Fe (III) complexes are powerful antioxidant agents due to their ability to scavenge superoxide free radicals and decrease the intracellular levels of reactive oxygen species (ROS). This could be attributed to the fact that iron binds more closely to RBP than other RBP–metal complexes [48]. The biological modification method alters the structure of polysaccharides by hydrolyzing the enzymes synthesized by microorganisms [32]. This may be due to the decrease in molecular weight of polysaccharides after fermentation. Similarly, modified rice bran polysaccharides (RBPSs) fermented using fungus *G. frondose* exhibited higher antioxidant activity than unfermented samples; after 9 days of fermentation, the DPPH scavenging ratio of polysaccharides (87.8%) at 2.0 mg/mL was significantly higher than that of unfermented polysaccharides (56.2%). HPLC analysis showed that the M_W_ of unfermented RBPSs was primarily between 10^3^ and 10^4^ Da, and later changed to 10^2^ and 10^3^ Da post-fermentation [47]. Meanwhile, Liu et al. [49] demonstrated that the modification of RBPSs using intracellular enzymes from *Grifola frondose* altered the molecular weight and monosaccharide composition of RBPSs, resulting in an increase in the DPPH scavenging ratio from 51.32% to 90.42%. The antioxidant activity was also enhanced in an in vivo model. RBP demonstrated antioxidation activity by significantly increasing the superoxide dismutase (SOD) and catalase (CAT) levels and decreasing MDA levels to a certain extent in the serum, liver, and spleen of mice. Further studies on the antioxidant mechanism showed that the expression of nuclear factor E2-related factor 2 was upregulated and the expression of NQ01 and HO-1 were downregulated, demonstrating the activation of the Nrf2/ARE pathway in the rice bran polysaccharide [36] (Figure 4).

### 4.2. Immunomodulatory Activity

Currently, natural polysaccharides are often used as pharmacological tools due to their powerful therapeutic potential [50]. Yamagishi et al. [42,51] confirmed that the rice bran water-soluble polysaccharide fractions showed similar or higher anti-complementary activity potency compared to water-soluble polysaccharides isolated from *Angelica acutiloba* and *Glycyrrhiza uralensis,* which have been proven to have better immunomodulatory effects. In an in vitro immunological test, Yi et al. [52] reported that RBP could significantly stimulate Concanavalin A-induced proliferation of spleen lymphocytes and enhance the phagocytosis of a macrophage when used in the dose range of 50~400 μg/mL. This indicated the immunopharmacological effect of RBP in vivo. Until now, researchers have put significant effort into establishing effective modification methods of natural polysaccharides to enhance biological activities. Cytokines modulate the immune response by binding to corresponding receptors to regulate cell growth, differentiation, and effects. In an immunological test, the highest immunological activity was obtained when the sulfation conditions included sulfur trioxide and the sample in a 2:1 (*w/w*) ratio, and the reaction was performed at 80 °C for 3 h. At this time, sulfated polysaccharides significantly upregulated the mRNA expression of cytokines (iNOS, TNF-a, IL-1β, and IL-10) compared to crude polysaccharides. [8]. Nitric oxide (NO) has the immunomodulatory effect of downregulating disease-related genes. Liu et al. [47] showed that *Grifola frondose*-modified RBP had a bidirectional effect on NO production after nine days of fermentation compared to unfermented polysaccharides. The cells cultured with 50–100 µg/mL of fermentation products showed reduced NO production, while higher concentrations of the fermentation products (200–400 µg/mL) stimulated NO production. Moreover, a crude polysaccharide from rice bran fermented using *Preussia aemulans* strongly stimulated macrophage proliferation (170.03 ± 9.64%) and NO production (15.01 ± 0.33 uM) [53]. Similarly, Yang et al. [31] also showed that rice hull polysaccharides (RHPS) exert immunomodulatory effects by enhancing the levels of natural antibodies (IgG, IgM, and IgA), cytotoxicity of splenic natural killer cells (NK cells), phagocytosis by macrophages, and induction of cytokines (IL-2, IFN-γ) (Figure 5).

### 4.3. Antitumor Activity

The number of cancer cases is increasing each year, and it has become one of the leading causes of death from non-communicable diseases worldwide. Approximately 9.96 million cancer deaths were recorded worldwide in 2020, as per the data published by the World Health Organization/International Agency for Research on Cancer [54,55]. Rice bran polysaccharides have gained much attention from medical and food researchers due to their potential biological activity, which may allow them to inhibit the growth of a variety of tumors (e.g., gastrointestinal tract cancers [56], mammary tumors [57], and Meth-A fibrosarcoma [58]). For instance, the rice bran polysaccharide RON (dextran-like a-glucan) exhibited antitumor activity against syngeneic tumors, Meth-A fibrosarcoma, and Lewis lung cancer at an optimal dose of approximately 30 mg/kg when administered intraperitoneally or orally [59]. Polysaccharides extracted using different approaches exhibited different biological activities due to different levels of purity, molecular weights (Mw), and structures [60,61]. Han et al. [14] experimentally concluded that compared with polysaccharides obtained using the hydrothermal method (Mw = 1.03 × 10^5^ g/mol) and microwave-assisted extraction (Mw = 2.62 × 10^5^ g/mol), RBP-E (Mw = 0.47 × 10^5^ g/mol) obtained via enzyme-assisted extraction had remarkably stronger antitumor activity against S180 cells in vitro. The inhibition ratios of RBP-H, RBP-M, and RBP-E on S180 cells were 33.2%, 42.4%, and 54.9%, respectively, indicating that the low Mw led to remarkable antitumor activity. Molecularly modified and structurally improved polysaccharides seem to have significant advantages in disease prevention [62]. Wang et al. [6] demonstrated that the antitumor activity of sulfated polysaccharides was significantly increased against human hepatoma and murine B16 melanoma when the sulfation degree was in the range of 0.81–1.29 and the carbohydrate content was in the range of 41.41–78.56%. Concurrently, it was also shown that sulfated rice bran polysaccharide SRBPS2a significantly inhibited the growth of mouse mammary tumors, both in vitro and in vivo. At 500 mg/mL, the inhibition rates of SRBPS2a and rice bran polysaccharide in the EMT-6 cells were 47.35% and 31.58%, respectively. Moreover, SRBPS2a increased the weight of the immune organs (thymus index, spleen index, and body weight of the mice). On the other hand, spectroscopy analysis using the FT-IR and ^13^C NMR confirmed C-2 and C-4 as the sulfate substitution sites without side chains [57]. Through further investigations, Wang et al. [37] demonstrated that the sulfated polysaccharide antitumor activity of SRBP2 could be attributed to the apoptosis process in tumor cells and the secretion of NO and tumor necrosis factor-α, which could be involved in the killing of tumor cells. Other research studies have employed biotransformation methods, especially fungal extracellular enzymes, to modify polysaccharides by reducing the macromolecular fragments of polysaccharides to small molecular fragments, thus facilitating its pharmacological effects in the body [63,64]. Li et al. [27] reported that the polysaccharides M-50-A (110 kDa) and A-50-G (14 kDa) from fermented rice bran exhibited a higher reduction in the proliferation rate of S180 sarcoma cells compared to natural rice polysaccharides (210 kDa). Similarly, GS-DRB polysaccharide from Ganoderma sinense defatted rice bran had a much greater inhibitory effect on H1299 non-small-cell lung cancer than GS-FRB from Ganoderma sinense full fat rice bran at different time points. This could be primarily due to the GSFPS-1: GSFPS-2 ratio, and the fact that GSFPS-2 with a smaller Mw had higher antitumor activity [65]. Furthermore, Lentinus edodes-modified rice bran polysaccharide RBEP inhibited the growth of sarcoma-180 cells and solid tumor-B16/Bl6 melanoma cells, both orally and intraperitoneally. On the other hand, intraperitoneal administration of RBEP was more effective than oral administration in inhibiting transplanted tumor cells; after transplanting Sarcoma-180 cells into the peritoneal cavity of the mice, the average survival period of mice in the RBEP group (50 mg/kg, intraperitoneal administration) was prolonged by 38% compared to the controls. Later, mice were transplanted with B16/Bl6 melanoma, and the tumor suppression rate in the RBEP group (50 mg/kg, intraperitoneal administration) was 41.7% compared to the control group [66].

### 4.4. DNA Delivery

Defective gene expression predisposes an individual to cancer, as well as genetic and non-genetic diseases. Gene therapy is an effective treatment for these types of diseases [67,68]. Polysaccharides are used as natural biological materials due to their good biocompatibility, biodegradability, and low toxicity [69]. Studies have reported that modified polysaccharides can lead to the aggregation of large proteins into dense structures that can be used as gene delivery vehicles [70,71]. For instance, the low-molecular-weight polyethyleneimine (PEI)-modified rice bran polysaccharide (PRBP) effectively compresses DNA, protects it from degradation by deoxyribonuclease I, and shows higher transfection efficiency compared to PEI [30]. Similarly, the polymer PRBP-TAT, formed by grafting PRBP with transcriptional activator protein (TAT), was less cytotoxic to human embryonic kidney cells, rat hepatocytes, and liver hepatocellular cells compared to PEI. On the other hand, it effectively protected against DNA degradation at low N/P ratios and showed a higher gene transfection efficiency than PEI and PRBP in vitro [9]. In another study by Liu et al. [72], PEI-modified rice bran polysaccharide-Fe (III) complex (PEI-PI), at a low weight ratio of 2 (vector/DNA), also protected DNA from degradation and showed more significant DNA aggregation ability and higher gene transfection compared to the PEI-modified RBP+Fe (III) system. At present, only a few studies have explored polysaccharides as gene-delivery vehicles. The aforementioned studies provide platform data for the research on other polysaccharides as biological materials, and promote the application and development of polysaccharides in biomedicine.

### 4.5. Antibacterial/Antiviral Activity

Viral and microbial infections are great threats to human health. However, most of the existing anti-viral treatments have side effects, such as toxicity or drug resistance. Therefore, green and safe natural antibacterial/viral ingredients have gradually become a research hotspot. Surin et al. [28] demonstrated that polysaccharides extracted from four different kinds of rice bran using hexane pretreatment inhibited the pathogenic bacteria *Staphylococcus aureus (S. aureus)* and *Escherichia coli (E. coli)*; the maximum clear zone diameter of *S. aureus* and *E. coli* and were 12.5 mm and 14.8 mm, respectively. In vivo tests revealed that mice injected intraperitoneally with α-glucan RBS showed enhanced resistance to *Listeria monocytogenes* and *E. coli,* as macrophage activity was enhanced [73].

Ghosh et al. [74] validated that sulfated rice bran glucans could be used as an entry-inhibiting anti-cytomegaloviral agent, specifically in the initial stage of virus invasion, suggesting that sulfate of rice bran dextran is an effective antiviral drug. Ray et al. [75] reported that the mean values of 50% inhibited sulfated dextrans, i.e., P444, P445, and P446 led inhibition of human cytomegalovirus (HCMV) with replication at a concentration of 2.44 ± 0.58, 2.52 ± 0.21, and 6.54 ± 0.21 μg/mL, respectively. These values were much lower than those of rice bran glucan S1G (12.48 ± 0.60 μg/mL), demonstrating that sulfated dextran exerted a more effective inhibitory effect on HCMV.

### 4.6. Hypoglycemic

The incidence of diabetes mellitus remains very high worldwide. Anti-diabetic drugs are widely used, but have certain side effects. Thus, polysaccharides, as a natural glucose-lowering substance, have become a research hotspot for experts and scholars. Hikino et al. [76] isolated a class of rice bran polysaccharides from rice bran and observed that these polysaccharides had significant hypoglycemic activity. Furthermore, Meng et al. [77] confirmed that rice bran polysaccharide could inhibit the activities of α-glucosidase and α-amylase, resulting in a reduction in the blood sugar level, which could be beneficial for treating type 2 diabetes.

### 4.7. Hypolipidemic

In addition to the pharmacological activities mentioned above, rice bran polysaccharides also play a certain role in lowering blood lipids. Hu et al. [78] showed that the hemicellulose extracted from rice bran could facilitate the excretion of cholesterol and bile acids, reducing the risk of cardiovascular disease. Further studies by Nie et al. [79] demonstrated that RBP administration significantly reduced body weight, liver weight, and adipose tissue in mice and lowered plasma levels of cholesterol (TC), triglyceride (TG), and low-density lipoprotein cholesterol (LDL-C) compared to mice on a high-fat diet. Additionally, it also demonstrated that oral administration of RBP exerted a hypolipidemic effect by regulating the expression of genes related to lipid metabolism.

## 5. Application

Rice bran resources are abundant, cheap, and easy to obtain. The development of rice bran polysaccharides is one of the most effective methods of deep processing and utilization of rice bran resources. Rice bran polysaccharides are beneficial for health and can be used in medicine, health products, beverages, food, and in other fields (Figure 6).

### 5.1. Food Field

Since rice bran is rich in nutrients and active ingredients, it could be used as an ingredient in various foods to enhance their flavor. Numerous studies have reported rice bran as a key ingredient in the production of cereal-based functional beverages [80,81]. Rice bran polysaccharides can be used to prepare high-fiber, low-fat baked goods to replace other dietary fiber sources such as oats, wheat flour, and wheat bran [82]. Further studies have shown that oxidized rice bran polysaccharides have enhanced water solubility and, thus, RBP could be used as a source of water-soluble dietary fiber for cookies and bread [83]. Patents in the World Intellectual Property Organization (WIPO) portal were searched using the term “rice bran polysaccharide” and this search revealed that the nutritional and functional properties of rice bran polysaccharides can also be used in cookies, beverages, cakes, puffed foods, and other food products [84,85,86,87].

### 5.2. Industry Field

Rice bran is rich in nutritional value and contains a high level of protein. It is often used as a supplementary energy source for animal feed in livestock and poultry farming. Surprisingly, rice bran polysaccharides have a larger role in the feed industry [88]. Zheng et al. [89] reported that polysaccharide-rich fermented rice bran extract (FBE) could improve broiler growth performance, increase bone strength, and improve immunity. They also studied the complementary effect of FBE on piglets’ intestinal health and growth, and reported that adding FBE to the diet improved growth by reducing the occurrence of diarrhea, as well as by reducing oxidative stress in the small intestine [90]. Additionally, rice bran polysaccharides could be used as fermentation liquids and emulsifiers in the industry [91,92]. Huang et al. demonstrated that modification of whey protein isolate solution (WPI) with rice bran polysaccharides enhanced the synergistic effect of both components and optimized the functional properties of WPI, such as emulsification, heat resistance, and acid resistance, thus resulting in efficient and stable new emulsifiers [93].

### 5.3. Medical Field

In addition, it is worth mentioning that the pharmacological activities of rice bran polysaccharides have been patented [94,95,96,97,98,99,100]. For instance, patent “CN109879980 Preparation method for rice bran polysaccharide metal complex, and antioxidant” and patent “CN111718971 Rice bran polysaccharide with lipid-lowering activity and preparation method of rice bran polysaccharide” showed the antioxidant and hypolipidemic therapeutic effects of rice bran polysaccharides [97,100]. Rice bran polysaccharides can also be used in the field of drug-carrying systems [101]. Interestingly, rice bran polysaccharides primarily exert their medicinal effects in the form of oral tablets, oral liquids, injections, and soluble powders [102,103,104,105].

### 5.4. Other Fields

Rice bran polysaccharides can also be used in tracer technology and additives in cosmetics. Liu et al. showed that the reducing ends of iron molecules in rice bran polysaccharide react with the active amino group in rhodamine B to prepare the fluorescently labeled rice bran polysaccharide iron, which provides a rapid and convenient method for tracing rice bran polysaccharide iron [106]. This study confirmed that the microemulsion additive made of carboxymethylated rice bran polysaccharide could be used as an important component of sunscreen, sunscreen cream, and other cosmetics, which can improve the sunscreen’s effect and, in turn, the skin’s antioxidant system, thus delaying aging [107].

## 6. Conclusions and Future Perspectives

As one of the important active substances of rice bran, rice bran polysaccharides possess high immunomodulatory, antioxidant, antitumor, and hypolipidemic properties. This indicates that rice bran polysaccharides have high commercial value and broad application prospects in the medicine, food, and chemical industries. To date, RBPs have exhibited the potential to meet application requirements in different areas. For instance, RBPs have gained much attention as a source of dietary fiber in food. In particular, RBPs have attracted increasing interest in the pharmaceutical field as gene delivery vehicles. Nevertheless, relative to its abundant sources, the practical utilization of RBP is currently very low. On the other hand, compared with other widely reported and used polysaccharides, such as the *Lentinus edodes* polysaccharide and the *Ganoderma lucidum* polysaccharide, research on RBP is still in its infancy and many issues regarding RBP research need to be addressed. Firstly, the extraction process of polysaccharides should be optimized. Extrusion treatment, bio-enzymatic methods, and ultrasonic methods could be used to synergistically extract rice bran polysaccharides to achieve a high yield of polysaccharides and lower the energy consumption in the extraction process. Meanwhile, new extraction methods in the future could enhance the functional activity of rice bran polysaccharides [53]. Secondly, the separation and purification processes of rice bran polysaccharides are tedious, and the yield is low. Thus, efficient separation and purification methods for rice brain polysaccharide production must be further investigated. Thirdly, the structural characterization of rice bran polysaccharides is currently focused on molecular weight and monosaccharide composition. With the advent of modern analytical techniques, integrated analysis using computer prediction with multiple analytical methods, such as static and dynamic light scattering, X-ray diffraction (XRD), circular dichroism (CD), atomic force microscopy (AFM), and scanning tunneling microscopy (STM), could provide a developmental direction for advanced structural analysis of polysaccharides [108]. Fourth, there is an urgent need to investigate the relationship between polysaccharide structure and biological activity through in-depth research of structural analysis and pharmacological experiments [109]. Fifth, there is a need for an in-depth study of the bioactive mechanisms of polysaccharides. Genomics, proteomics, and metabolomics could be used to study the effects of polysaccharides on cell- or animal-related genes, proteins, and metabolites to gain insight into their bioactive mechanisms [110].

## Figures and Tables

**Figure 1 foods-12-00639-f001:**
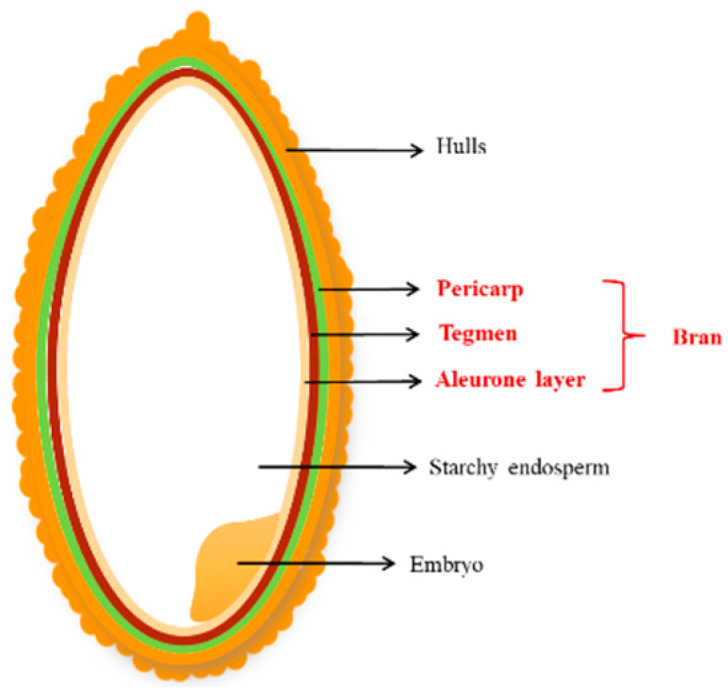
The anatomy of a rice grain.

**Figure 2 foods-12-00639-f002:**
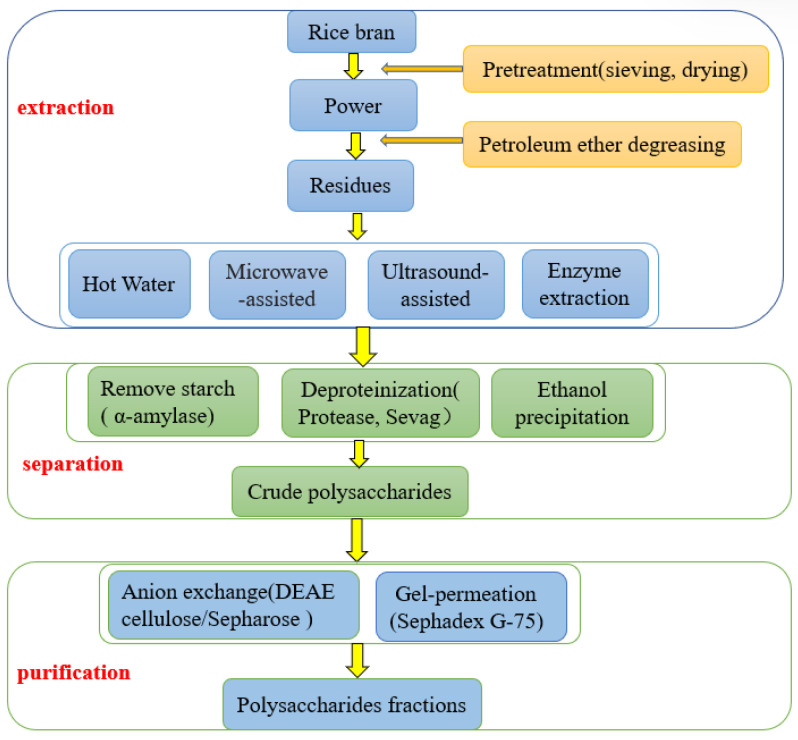
The process of extraction, separation, and purification of bran rice polysaccharides.

**Figure 3 foods-12-00639-f003:**
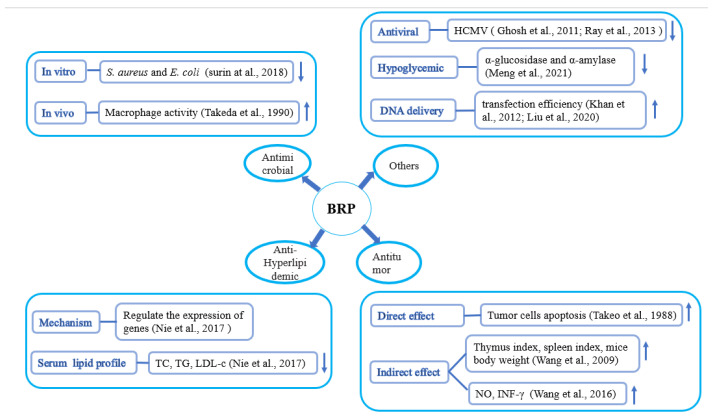
Bioactivities of rice bran polysaccharides.

**Figure 4 foods-12-00639-f004:**
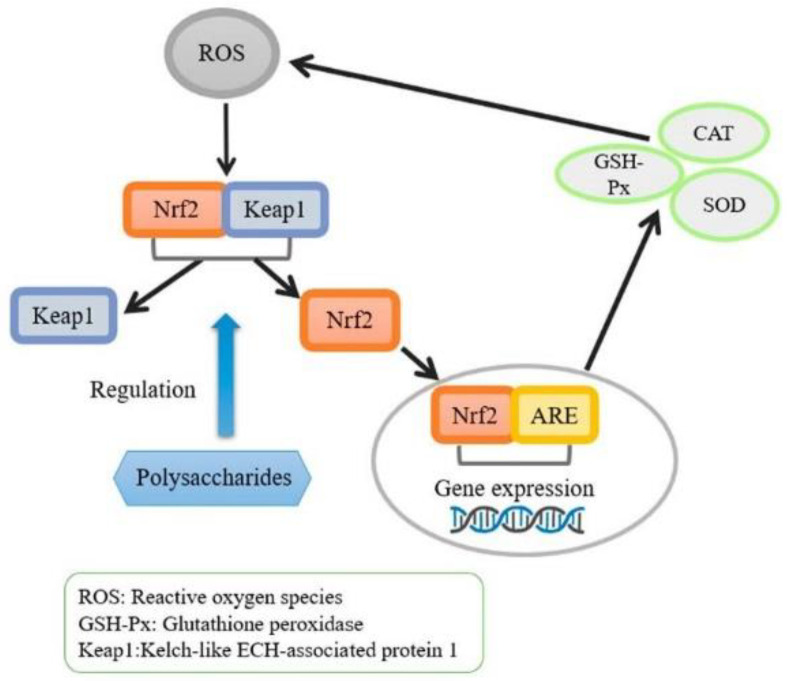
Signaling pathway of antioxidant activity.

**Figure 5 foods-12-00639-f005:**
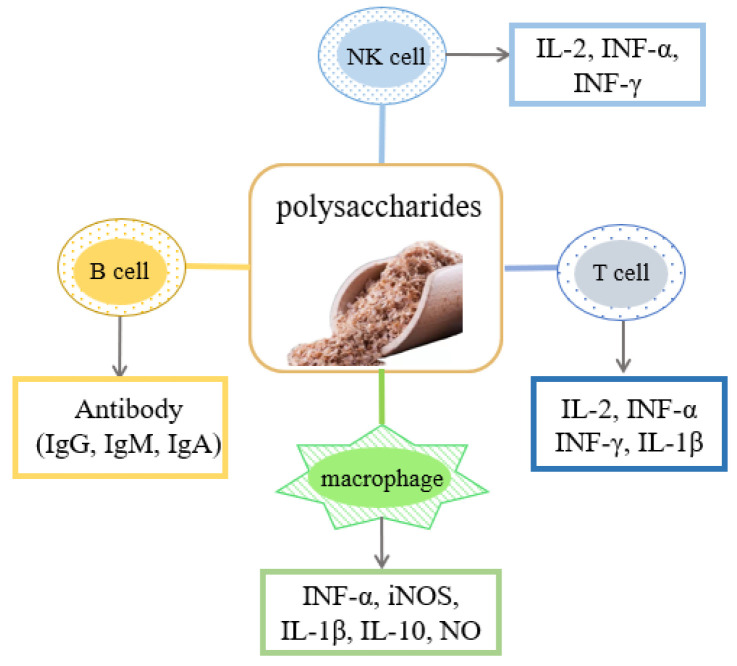
Main mechanisms of action of immunomodulatory activity.

**Figure 6 foods-12-00639-f006:**
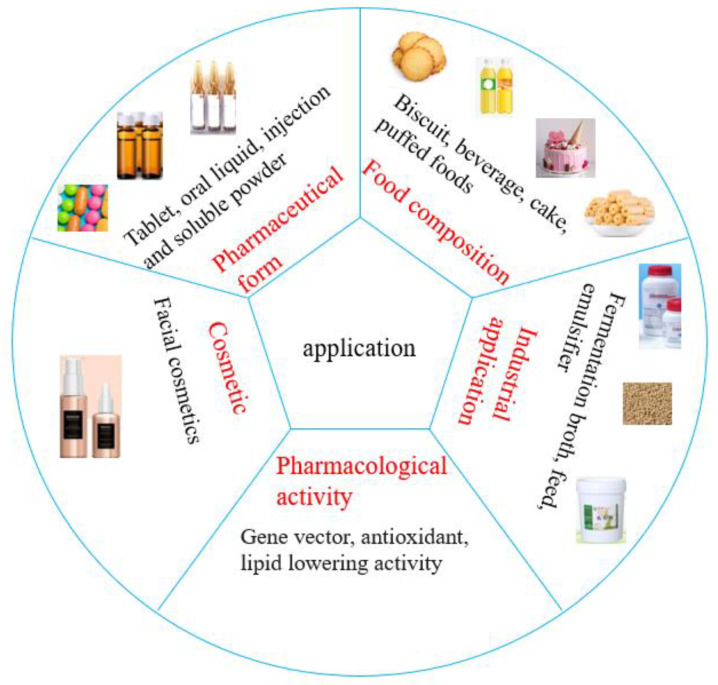
The application of rice bran polysaccharides.

**Table 1 foods-12-00639-t001:** Structural characteristics of the rice bran polysaccharides.

Polysaccharide Fraction	Extraction, Separation, and Purification Procedures	Mw (KDa)	Monosaccharide Composition	Structure Features	Structural Characterization Method	Reference
RBP	Hot water extraction, DEAE-52 ion exchange chromatography, dextran gel tomography	488	Glu:Xyl:Gal:Ara = 1.669:1:0.225:1.228	α-D-glycoside bond.	HPLC, FTIR, NMR,	[30]
RHPS	Hot distilled water (90–95 °C, 5 h), DEAE column	77	Ara:Gal:Glu:Man:Xyl = 10:44.8:29.8:9.3:6.1	(1 → 3)-Gal as backbone	HPAEC-PAD, HP-SEC, NMR, GC-MS	[31]
M-50-A	Hot water extraction, DEAE Sephadex A-25 anion-exchange and Sephadex G-75 gel-permeation chromatography	110	Ara:Xyl:Glu:Man: Gal = 1.7:7.2:1:11:1.4	α-D-pyranoid polysaccharide and alpha-residues	HPGPC, GC, FT-IR	[32]
A-50-G	Hot water extraction, DEAE Sephadex A-25 anion-exchange and Sephadex G-75 gel-permeation chromatography	14	Ara:Xyl:Glu:Man: Gal = 1.2:1.5:1:11.1:2.4,	α-D-pyranoid polysaccharide and alpha-residues	HPGPC, GC, FT-IR	[32]
RBP1, RBP2 and RBP3	Distilled water (50–60 °C, 1 h) DEAE-Cellulose and Sephadex G-200 column chromatography	51.5, 26.1 and 6.95			GPC, GC-MS	[34]
RBPS2a	Hot water extraction	90	Ara:Xyl:Glu:Gal = 4:2:1:4	β-(1 → 3)-linked d-galactopyranosyl residues substituted at O-2 with glycosyl residues composed of α-d-xylose–(1 → 4)-α-d-arabinose (1 → and α-d-glucose–(1 → 4)-α-d-arabinose (1 → linked residues	HPGPC, GC, FT-IR, NMR	[35]
Rice bran polysaccharide	Hot distilled water (100 °C, 5 h), Sephadex G100 column chromatography			α-1,4 and α-1,6 glycosidic bonds	FT-IR, NMR	[36]
RBP2	Water extraction (90 °C, 2 h)		Gal:Ara:Rha:Glu:Xyl:Man = 41.51:22.58:12.89:9.56:7.01:3.75	Arabingalactan in its main chain and a xylose polymer in its side chain	GC, NMR	[37]

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
