# Peer review of "Extraction, Structural Characterization, Biological Functions, and Application of Rice Bran Polysaccharides: A Review"

_foods, 2023, doi:10.3390/foods12030639_

Round 1

Reviewer 1 Report

This manuscript is about a review of rice bran polysaccharides, and I have some considerations:

- I suggest including a figure indicating where is the rice bran in rice grain

- The references are missing in the end of archive

- I suggest more development in the discussion of polysaccharide extraction. For example, the use of ultrasound or microwave altered the polysaccharide sturcture? Promote degradation? Was there any study using pressurized water extraction? 

- IN application, is there any study in food industry with polysaccharide or oligosaccharides?

Author Response

Thanks for your careful read and thoughtful comments concerning our manuscript. The corrections in the paper and the responds to your comments are as flowing:

Comments 1: I suggest including a figure indicating where is the rice bran in rice grain.

Response: Thank you for your suggestion. According to the reviewer's suggestion, we added an anatomy of the rice grain in the Figure 1 and marked the location of the rice bran in the grain.

Comments 2: The references are missing in the end of archive.

Response: Thank you for your comments. Corresponding references 108-111 have been added in the end of archive.

Comments 3: I suggest more development in the discussion of polysaccharide extraction. For example, the use of ultrasound or microwave altered the polysaccharide structure? Promote degradation? Was there any study using pressurized water extraction?

Response: We are extremely grateful to reviewer for pointing out this problem. We analyzed the effects of ultrasound and microwave on the structure of polysaccharide in 2.1. of the manuscript. Since there are no studies on pressurized water extraction of rice bran, there are no relevant aspects to add.

Comments 4: In application, is there any study in food industry with polysaccharide or oligosaccharides?

Response: We deeply appreciate the reviewer’s suggestion. At present, there are many studies on the application of rice bran polysaccharide in food (5.1. of the text), but there is no industrialized product yet, and it is necessary to continue to study in depth.

Reviewer 2 Report

The manuscript entiled: ”Extraction, structural characterization, biological functions, and industrial application of rice bran polysaccharides: A review” reports information on rice bran polysaccarides from different perspective points of view. Even if interesting, the manuscript proposed lacks of novelty and adds limited contribution to the area of interest considering the existing literature focused on the topic of interest. There are also some drawbacks, like the use of many abbreviations which remain undefined (see e.g. “RHPS” in Table 1, etc. Figure 2 should be commented in the text and the different activities mentioned better exploited and substantiated. Data would be needed to support the cited propertis with reference to the cited literature data. Overall the manuscript would need a better assessment for a better readability.

Author Response

Thanks for your careful read and thoughtful comments concerning our manuscript. The corrections in the paper and the responds to your comments are as flowing:

Comments 1: There are also some drawbacks, like the use of many abbreviations which remain undefined (see e.g. “RHPS” in Table 1, etc. Figure 2 should be commented in the text and the different activities mentioned better exploited and substantiated. Data would be needed to support the cited properties with reference to the cited literature data. Over all the manuscript would need a better assessment for a better readability.

Response: We deeply appreciate the reviewer’s suggestion. To be more clearly and in accordance with the reviewer concerns, we have added a detailed explanation of “RHPS”. In addition, Figure 2 (Now it changes to Figure 3) is annotated with references to better utilize and confirm the mentioned activities, also, relevant data have been added to support the corresponding activities (4.1., 4.2., 4.3., 4.5. of the text).

Reviewer 3 Report

Rice is a staple across Asia, South America, and Africa. The rice industry's main byproduct, rice bran, is removed from brown rice during milling. Compared to white rice, brown rice provides more protein and fiber. It makes up 5%–8% of a rice grain's weight and includes the exocarp, mesocarp, cross-linked layer, aleurone layer, and other layers. The world's 500 million tons of rice produce 25 million tons of rice bran. Rice bran's nutritional content is impressive and readily available. Nutritional breakdown of rice bran: Protein, fat, and fiber: 12%-16%, 12%-23%, and 23%-30%, respectively. Glutamate, phenanthrene, inositol, and phytic acid are also present. According to ancient Chinese medicinal books, rice bran may heal foot illnesses by nourishing the bowels, boosting the appetite, and tonifying the qi and blood. Due to this, many call it a "treasure house of natural nourishment." Sadly, most rice bran is wasted as poultry feed or dumped. According to the UN Industrial Development Organization, rice bran is one of the natural raw resources that is underutilized (UNIDO).

Polysaccharides, a primary functional component of rice brans, may prevent tumor development, protect cells from free radical damage, and affect the immune system, according to recent studies. Rice bran polysaccharide (RBP) may be used in gene and drug delivery. RBP's biological activities have garnered notice lately. As of this writing, just one review article on shiitake mushroom enzyme-modified rice bran arabinoxylan compound (RBAC) is accessible. The authors have summarized current work on rice bran polysaccharides' extraction, structural characteristics, biological significance, and possibilities for future use (with the exception of the RBAC).

Though the paper is well-written, I would like to see the following changes

·         The authors should provide schematic diagrams for each of the pharmacological activities.

·         The application section should be divided into sub-sections (namely, pharmaceutical formulations, food products, industrial applications, and cosmetics) and elaborated.

·         The physicochemical analysis of the RBP should be included in the review.

Author Response

Thanks for your careful read and thoughtful comments concerning our manuscript. The corrections in the paper and the responds to your comments are as flowing:

Comments 1: The authors should provide schematic diagrams for each of the pharmacological activities.

Response: We are extremely grateful to reviewer for pointing out this problem. and we have added schematic diagrams of antioxidant (Figure 4) and immunomodulatory (Figure 5) activity mechanisms to the original activity schematic. Since the current research on other activities is relatively shallow and the mechanism is not very clear, we are very sorry that we can only give a brief description based on the existing research.

Comments 2: The application section should be divided into sub-sections (namely, pharmaceutical formulations, food products, industrial applications, and cosmetics) and elaborated.

Response: In accordance with the reviewers' comments, we have divided the application section into subsections (namely, food field (5.1.), industry field (5.2.), medical field (5.3.), and other fields (5.4.)) and elaborated on them.

Comments 3: The physicochemical analysis of the RBP should be included in the review.

Response: Thank you for the suggestion. We have added physicochemical analysis of the RBP in the review (3.1.,3.2.,3.3. of the text).

Round 2

Reviewer 2 Report

The manuscript yet modified does not evidence the novelty and seems to add limited contribution to the area of interest. This aspect should be evidenced in a review manuscript. Moreover these aspects have been already put in evidence in the first Referee round (please check previous comments) and have not been comments or addressed in the reply by the Authors.

Author Response

Comments 1: The manuscript yet modified does not evidence the novelty and seems to add limited contribution to the area of interest. This aspect should be evidenced in a review manuscript. Moreover these aspects have been already put in evidence in the first Referee round (please check previous comments) and have not been comments or addressed in the reply by the Authors.

Response: We are very sorry for not responding to the question about novelty due to our negligence. We think the novelty of the article is mainly reflected in the following points. Firstly, till now, there are no systematic study on the extraction and purification techniques for RBP and the structural characterization and pharmacological effects of RBP, this review aims to stimulate the interest of researchers and provide valuable resources and information. Secondly, the research on rice bran polysaccharides as gene delivery vehicles could provide platform data for the research on other polysaccharides as biological materials and promote the application and development of polysaccharides in biomedicine. Finally, this review focuses on the applications of rice bran polysaccharides in food, industrial, and pharmaceutical, which is helpful for the subsequent development and utilization of rice bran polysaccharides.

We deeply appreciate the reviewer’s suggestion and realize that these differences may not have been expressed clearly enough in the previous manuscript. We have made improvements to the original manuscript, particularly in the introduction (paragraph 2) and discussion, in order to state the purpose of the article study clearly.

Reviewer 3 Report

Figure 2.: "Sievding" should be replaced with "Sieving"

Author Response

Comments 1: Figure 2.: "Sievding" should be replaced with "Sieving"

Response: In accordance with the reviewers' comments, we have replaced "Sievding" with "Sieving" (Figure 2.)